# A defined syphilis vaccine candidate inhibits dissemination of *Treponema pallidum* subspecies *pallidum*

Karen V. Lithgow[1,*], Rebecca Hof[1,*], Charmaine Wetherell[1], Drew Phillips[1], Simon Houston[1] & Caroline E. Cameron[1]

Syphilis is a prominent disease in low- and middle-income countries, and a re-emerging public health threat in high-income countries. Syphilis elimination will require development of an effective vaccine that has thus far remained elusive. Here we assess the vaccine potential of Tp0751, a vascular adhesin from the causative agent of syphilis, *Treponema pallidum* subsp. *pallidum*. Tp0751-immunized animals exhibit a significantly reduced bacterial organ burden upon *T. pallidum* challenge compared with unimmunized animals. Introduction of lymph nodes from Tp0751-immunized, *T. pallidum*-challenged animals to naive animals fails to induce infection, confirming sterile protection. These findings provide evidence that Tp0751 is a promising syphilis vaccine candidate.

[1] Department of Biochemistry and Microbiology, University of Victoria, 3800 Finnerty Road, Victoria, British Columbia V8P 5C2, Canada. * These authors contributed equally to this work. Correspondence and requests for materials should be addressed to C.E.C. (email: caroc@uvic.ca).

Syphilis, caused by infection with the spirochete *Treponema pallidum* subsp. *pallidum* (hereafter referred to as *T. pallidum*), remains a prevalent disease, particularly in low- and middle-income countries, with an estimated global burden of 36 million cases and 11 million new cases per year[1]. Rates of infectious syphilis among men who have sex with men (MSM) are exhibiting an alarming increase, with a rate of 228.8 cases per 100,000 MSM population (0.228%) reported in the United States in 2013 compared with a rate of 15.8 per 100,000 MSM population (0.016%) reported in 2000 (ref. 2). Similar increases are being observed in China[3], England[4], Canada[5] and Australia[6]. Mother-to-child transmission of syphilis is also on the rise, with an estimated 1.36 million pregnant women infected worldwide each year and over 500,000 of these pregnancies leading to adverse outcomes including spontaneous abortion, stillbirth, premature delivery, neonatal death and manifestations of congenital syphilis[7]. Symptomatic syphilis infections increase human immunodeficiency virus (HIV) transmission and acquisition two- to fivefold, and modelling studies predict that eradication of syphilis would have a significant impact on HIV prevention[8,9]. The increasing prevalence of infectious and congenital syphilis, despite the continued sensitivity of *T. pallidum* to treatment with penicillin, underscores the need for syphilis vaccine development as a complement to traditional screening and treatment approaches for the global elimination of syphilis. In addition, although *T. pallidum* has not yet exhibited resistance to penicillin, resistance to second-line therapy such as macrolide-based antibiotics has emerged[10]. As a result, and because of the devastating consequences of congenital syphilis, penicillin-allergic pregnant women require desensitization and treatment with penicillin, further emphasizing the need for an alternative control measure.

Effective syphilis vaccine development needs to target the highly invasive nature of *T. pallidum* that disseminates via the bloodstream and lymphatics and invades a wide variety of tissues and organs[11]. *Treponema pallidum* subsp. *pallidum* crosses endothelial, placental and blood–brain barriers early in infection, as demonstrated by the widespread clinical manifestations associated with syphilis infections, the

occurrence of congenital syphilis and the central nervous system invasion observed in ∼40% of early syphilis patients[11,12]. However, there is a limited understanding of the mechanisms responsible for the widespread dissemination capability of *T. pallidum*.

One *T. pallidum* protein that has been implicated in treponemal dissemination is the adhesin Tp0751. Opsonophago-cytosis assays provide evidence for the presence of this lipoprotein on the *T. pallidum* surface, which would facilitate direct interaction with the host environment[13]. Tp0751 has been demonstrated to bind to multiple extracellular matrix components, including laminin[14], fibronectin and fibrinogen[15]. All of these host components are in close proximity to the vascular endothelium, and therefore interaction of Tp0751 with these molecules is suggested to ideally position *T. pallidum* to facilitate dissemination via the bloodstream.

Previous attempts at vaccine development for syphilis have achieved varying degrees of success[16]. Immunization of rabbits, the optimal animal model that recapitulates the multiple stages and chronicity of human syphilis, with a selection of individual recombinant proteins has elicited at best partial protection[17–30]. More promising protection results have been achieved using attenuated whole-cell *T. pallidum* preparations[31–33]. As reported by Miller[33] in 1973, immunization with γ-irradiated *T. pallidum* provided complete protection upon homologous challenge in the rabbit model, as demonstrated by the absence of primary lesions at challenge sites and the failure to induce infection upon transfer of lymph nodes from these rabbits to naive rabbits. In the Miller[33] study, rabbits were given 60 intravenous injections containing a total of $3.7 \times 10^9$ γ-irradiated *T. pallidum* over a 37-week period. Although this immunization regimen is impractical for use in humans, this study significantly advanced the field of syphilis vaccine development as it illustrated that sterile protection can be successfully achieved with appropriate antigen selection and, simultaneously, it highlighted the importance of *T. pallidum* surface antigens in conferring protection[33].

Here we investigate the capacity of Tp0751 immunization to inhibit dissemination of *T. pallidum* to distant organ sites in rabbits. We show that animals immunized with Tp0751 display a

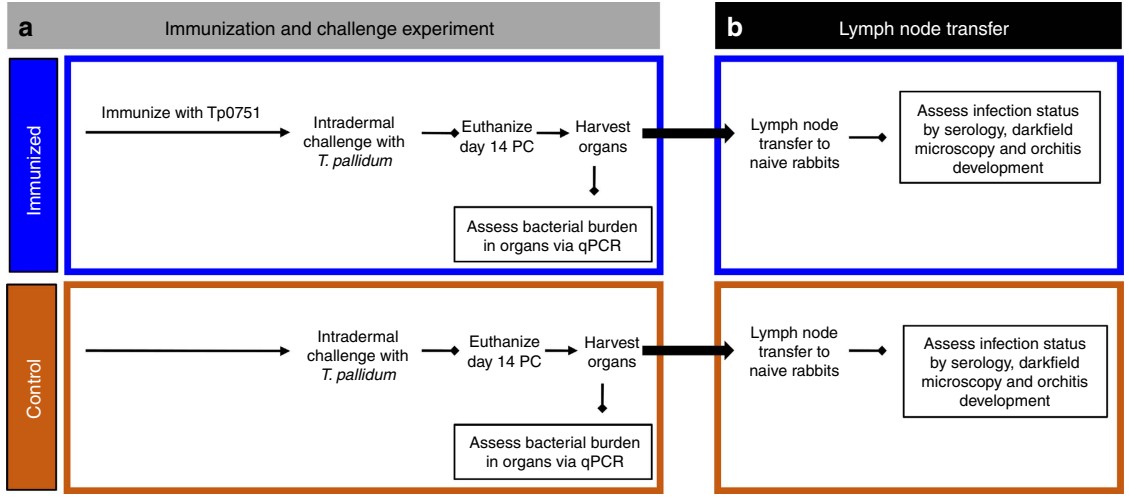

**Figure 1 | Experimental overview.** (**a**) Immunization and challenge. Recombinant Tp0751-immunized rabbits ($N = 3$; each immunization was delivered as four 0.1 ml subcutaneous (SC) injections into the shoulders and two 0.04 ml intramuscular (IM) injections into the quadricep muscles) and control animals ($N = 2$) were intradermally challenged with $1 \times 10^6$ *T. pallidum* subsp. *pallidum*, Nichols strain, at each of 10 sites. On day 14 post challenge (PC), animals were euthanized and organs were harvested for qPCR assessment of bacterial burden; popliteal lymph nodes were extracted and processed for the lymph node transfer. (**b**) Lymph node transfer. Popliteal lymph nodes were collected from recombinant Tp0751-immunized and control animals and injected into the testes of naive animals. These animals were monitored for evidence of infection using serology, darkfield microscopy and orchitis development over 185 days.

reduced treponemal burden in multiple organs distant to the challenge site as compared with control unimmunized animals. In addition, we demonstrate that lymph nodes recovered from Tp0751-immunized animals fail to induce a productive infection when injected into naïve recipient animals, further demonstrating that Tp0751 immunization inhibits treponemal dissemination to distant organ sites. Our results support Tp0751 as a promising vaccine candidate for preventing haematogenous spread of *T. pallidum* within the host.

## Results

**Tp0751 immunization attenuates lesion development**. Three rabbits were sedated and immunized three times with Tp0751 emulsified in TiterMax Gold adjuvant at 3-week intervals, resulting in induction of high titres of anti-Tp0751 antibodies in each of the immunized animals (Fig. 1 and Supplementary Figs 1 and 2). Previous control investigations conducted using an adjuvant-only immunization regimen failed to show a protective effect from mock immunization with a rabbit adjuvant in the absence of antigen[23], thus ruling out the possibility that adjuvant alone confers protective immunity. At 3 weeks following the final immunization, the immunized animals and the control unimmunized animals were sedated and intradermally challenged with $1 \times 10^6$ *T. pallidum* subsp. *pallidum*, Nichols strain, at each of 10 sites on their shaved backs (total of $1 \times 10^7$ *T. pallidum*/rabbit; Fig. 1a). During the subsequent 14 days, the challenge sites were monitored for the appearance of the painless lesions, or chancres, that arise at the site of inoculation and are unique to syphilis[11]. The chancre represents the first clinical sign of infection, and therefore monitoring chancre development provides a metric of vaccine efficacy. Lesions were monitored daily for diameter, induration, ulceration and presence of viable *T. pallidum*. By day 14, lesions were present on all animals; however, immunized animals exhibited delayed lesion development as compared with the unimmunized control animals. Control animals presented 100% lesion induration (skin hardening because of inflammation) by day 4, whereas immunized animals only reached 67% of the induration seen in control animals by day 4 and did not match the level of induration observed in the control animals until day 10 (Table 1). Painless ulceration in the control animals was apparent as early as day 10 and 100% of lesions were ulcerated by day 14. Immunized animals displayed signs of painless ulceration by day 10, but by day 14 only 47% of lesions were ulcerated (Fig. 2a). The lesion diameter observed in control animals was consistently 20% larger than that in immunized animals over the 14-day measurement period (Fig. 2b and Table 1). At day 10 post challenge (PC), each lesion was evaluated by darkfield microscopy for the presence of viable *T. pallidum*. 70% of lesions from control animals were positive for *T. pallidum* as compared with only 20% of lesions from immunized animals (Fig. 2c, $P = 0.002$, Student's two-tailed *t*-test). Furthermore, darkfield-positive lesions from controls had a significantly higher density of treponemes and at least 50% of these organisms were motile, whereas darkfield-positive lesions from immunized animals had only a few treponemes present per field and these organisms were non-motile. Analysis of lesion progression revealed a significant decrease in lesion ulceration (Fig. 2a, $P = 0.0011$) and treponemal burden (Fig. 2c, $P = 0.02$), and a trend towards decreased lesion diameter (Fig. 2b, $P = 0.11$) in Tp0751-immunized animals.

**Tp0751 immunization inhibits *T. pallidum* dissemination**. To evaluate whether Tp0751 immunization confers protection against treponemal dissemination, control and immunized animals were euthanized 14 days PC. We then performed quantitative real-time PCR (qPCR) to assess the *T. pallidum* burden in biopsy samples from lesions arising at the challenge sites and organ sites distant to the location of challenge. Analysis of the average local treponemal burden revealed a slight trend towards increased *T. pallidum* DNA concentrations (*flaA* copies per μg host tissue DNA) at the primary lesion sites of the three immunized animals compared with the two control unimmunized animals (Fig. 3a). However, it should be noted that the higher average burden resulted from detection of a higher treponemal burden in immunized rabbit 1 (Im1), whereas rabbits Im2 and Im3 exhibited lower treponemal burdens than the control animals (Fig. 3a and Table 1). Somewhat contradictory to this observation, microscopic assessment of primary lesions revealed that all immunized animals harboured lower treponemal burdens compared with the control group (Fig. 2c). Whereas darkfield analysis of lesion aspirates from infected animals allows for differentiation between live and dead organisms based upon motility, qPCR detects the DNA from both live and dead organisms that would, predictably, result in a higher value. A detailed assessment of the viability of treponemes in Tp0751-immunized animals PC, likely using reverse transcriptase-qPCR analyses to detect RNA in viable organisms, must await further studies.

Treponemal burden was also assessed at the distant organ sites of the bone, liver and spleen to provide an indication of the capacity of *T. pallidum* to disseminate from the site of initial infection. These sites were chosen because of the propensity for *T. pallidum* to disseminate to, and be detected in, these organs[34]. Treponemal burden in bone, liver and spleen extracts revealed a generalized trend towards lower DNA concentrations in all immunized animals (Fig. 3b–d). The most prominent decrease was observed for *T. pallidum* DNA concentrations in the spleen of the immunized animals compared with the nonimmunized controls (Fig. 3d). Importantly, analysis of the total treponemal burden in all distant infection sites revealed a significant decrease in the burden within immunized animals relative to the nonimmunized controls (Fig. 3e, $P < 0.005$, Mann–Whitney test), indicating that Tp0751 immunization inhibits *T. pallidum* dissemination to these distant organ sites.

To further determine whether Tp0751 immunization protects against *T. pallidum* dissemination, popliteal lymph nodes were transferred from two control and three immunized animals into five naïve, unexposed recipient animals (Fig. 1b). The infection status of recipient animals was assessed by serology (via the venereal disease research laboratory (VDRL) and fluorescent treponemal antibody-absorption (FTA-ABS) tests), darkfield microscopic analysis of testicular aspirates and testes orchitis development (Fig. 1b and Supplementary Table 2). In line with the trend observed in the qPCR results, transfer of popliteal lymph nodes from the control unimmunized animals to naïve recipient animals resulted in evidence of seroconversion by day 17 post transfer, strong seroconversion by day 31 post transfer and development of asymptomatic orchitis by day 42 post transfer (Table 2 and Supplementary Table 2). Darkfield analysis of the testes of control recipient animals at the time of euthanasia confirmed the presence of motile *T. pallidum*. Conversely, none of the three naïve animals receiving lymph nodes from the Tp0751-immunized animals developed orchitis; two of the animals receiving lymph nodes from immunized animals (designated R-Im1 and R-Im2) did not seroconvert during the duration of the experiment, and at least two of the recipient animals (R-Im1 and R-Im2) did not have treponemes present via darkfield analysis (Table 2; darkfield analysis not performed for R-Im3).

Intriguingly, rabbit Im3 had the lowest treponemal burden detected in the lesion biopsies, with an average of $1.66 \times 10^6$ *flaA*

**Table 1 | Experimental data for individual control and immunized animals.**

| Immunization and challenge experiment | | | Control | | Immunized | | |
|---|---|---|---|---|---|---|---|
| | | | Ct1 | Ct2 | Im1 | Im2 | Im3 |
| Lesion status | Diameter (mm)* | | $14.53 \pm 2.00$ | $16.01 \pm 2.00$ | $12.48 \pm 1.00$ | $12.13 \pm 1.00$ | $11.30 \pm 1.00$ |
| | Ulceration (%)* | | 100 | 100 | 40 | 30 | 70 |
| | Induration[†] | | 4 | 4 | 10 | 10 | 10 |
| *T. pallidum* burden | Darkfield analysis (%)[‡] | Lesion | 77.8 | 60 | 30 | 0 | 20 |
| | qPCR[§] | Lesion | $4.3 \times 10^6 \pm 5.2 \times 10^5$ | $7.1 \times 10^6 \pm 1.8 \times 10^6$ | $2.0 \times 10^7 \pm 6.6 \times 10^6$ | $3.2 \times 10^6 \pm 8.2 \times 10^5$ | $1.66 \times 10^6 \pm 7.6 \times 10^5$ |
| | | Bone | $1.28 \times 10^2 \pm 6.5 \times 10^1$ | $2.98 \times 10^2 \pm 6.1 \times 10^1$ | $0 \pm 0$ | $2.36 \times 10^2 \pm 4.4 \times 10^1$ | $0 \pm 0$ |
| | | Liver | $5.20 \times 10^3 \pm 3.0 \times 10^3$ | $2.47 \times 10^3 \pm 6.2 \times 10^2$ | $5.85 \times 10^2 \pm 5.9 \times 10^2$ | $1.11 \times 10^3 \pm 3.6 \times 10^2$ | $1.81 \times 10^2 \pm 1.8 \times 10^2$ |
| | | Spleen | $1.91 \times 10^3 \pm 2.4 \times 10^2$ | $1.55 \times 10^3 \pm 7.4 \times 10^1$ | $7.11 \times 10^2 \pm 1.9 \times 10^2$ | $9.16 \times 10^2 \pm 8.9 \times 10^1$ | $2.34 \times 10^2 \pm 1.2 \times 10^2$ |
| Cellular infiltration | Histology[∥] | Neutrophils | $2.2 \pm 0.3$ | $1.2 \pm 0.3$ | $6.0 \pm 0.7$ | $3.1 \pm 0.4$ | $7.1 \pm 1.1$ |
| | | Dendritic cells | $6.5 \pm 0.8$ | $5.4 \pm 0.9$ | $3.5 \pm 0.3$ | $4.3 \pm 0.9$ | $13.9 \pm 1.8$ |
| | | Lymphocytes | $24.1 \pm 3.0$ | $25.8 \pm 2.7$ | $2.1 \pm 0.4$ | $39.7 \pm 5.4$ | $55.6 \pm 8.7$ |
| | | Differentiated B cells | $1.6 \pm 0.4$ | $1.2 \pm 0.3$ | $1.2 \pm 0.3$ | $3.9 \pm 0.8$ | $4.9 \pm 0.6$ |

*Mean ± s.e.m. from 10 lesions at day 14 post challenge.
[†]Days post challenge to reach 100% induration.
[‡]At day 10 post challenge.
[§]Mean ± s.e.m. *flaA* copy number per µg rabbit DNA at day 14 post challenge.
[∥]Mean ± s.e.m. cell count per field of view (× 400 magnification) at day 14 post challenge.

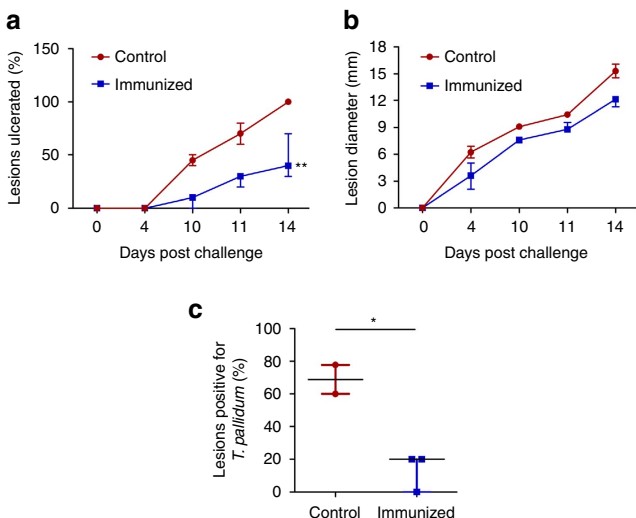

**Figure 2 | Immunization with Tp0751 decreases severity of and bacterial load in primary lesions.** (**a**) Lesion ulceration and (**b**) lesion diameter were monitored over 14 days in control ($N = 2$) and immunized ($N = 3$) animals following *T. pallidum* challenge. (**c**) Bacterial load in lesions was assessed at day 10 PC by examining the presence of *T. pallidum* using darkfield microscopy. Results are presented as median ± 95% confidence interval between individual immunized ($N = 3$) or control ($N = 2$) animals. Significance was assessed using nonlinear regression and extra sum of squares *F*-test (**$P < 0.005$ (**a**,**b**)) or a two-tailed *t*-test (*$P < 0.05$ (**c**)).

copies per µg of rabbit DNA detected compared with $2.04 \times 10^7$ and $3.15 \times 10^6$ *flaA* copies per µg of rabbit DNA for rabbits Im1 and Im2, respectively (Fig. 3a and Table 1). This may be suggestive of decreased proliferation of *T. pallidum* at lesion sites or may indicate a greater number of *T. pallidum* disseminated from the site of challenge in this rabbit. In line with this observation, lymph node transfer from rabbit Im3 to naïve rabbit R-Im3 led to seroconversion within this animal at day 73, suggesting that some treponemes may have been present in the transferred lymph nodes, resulting in a low-level infection that led to late seroconversion (Table 1 and Supplementary Table 2). However, the treponemal burdens detected in the spleen and liver of rabbit Im3 were the lowest of the three immunized animals (Fig. 3c,d, Table 1), suggesting the seroconversion observed in

rabbit R-Im3 did not occur as a result of increased treponemal dissemination to distant organ sites in rabbit Im3. Supporting evidence for this comes from the weak reactivity of the VDRL that did not increase significantly over time, and the slow and minimal increase of the FTA-ABS observed for rabbit R-Im3 (Table 1 and Supplementary Table 2).

**Immunization with Tp0751 promotes cellular infiltration.** Biopsy samples obtained from primary lesion sites of immunized and control animals were analysed for the presence of four immune cell subsets. Immunized animals had higher levels of cellular infiltrates in all cell types analysed, including neutrophils, dendritic cells, lymphocytes and differentiated B cells (Fig. 4). Analysis of the total cellular infiltrate composition in primary lesion sites revealed a significant increase in the amount of cellular infiltration in lesions from Tp0751-immunized animals versus nonimmunized controls (Fig. 4e, $P = 0.045$, Mann–Whitney test). Notably, immunized rabbit Im3 had the highest cell counts for every type of immune cell analysed, suggesting that this animal had the most pronounced immune cell infiltration at day 14 PC (Fig. 4, Table 1). Consistent with this observation, rabbit Im3 also had the lowest treponemal burden of all animals in the local lesion site as well as the disseminated infection sites. These included the liver and bone, where no treponemal DNA was detected by qPCR analysis (Fig. 3b,c and Table 1), as well as the spleen, where a low level of treponemal DNA was detected (Fig. 3d and Table 1).

**Discussion**
The results presented in this study demonstrate that animals immunized with Tp0751 display attenuated lesion development, inhibition of *T. pallidum* dissemination and increased cellular infiltration at lesion sites (Fig. 5a). Treponemal burden in primary lesion sites was analysed both visually by darkfield microscopy and quantitatively via qPCR. Compared with the two unimmunized controls, all three immunized animals displayed attenuated lesion development and, by darkfield analysis, significantly lower burdens of *T. pallidum* at the primary lesion sites. However, qPCR results demonstrated that immunized rabbit Im1 had the highest treponemal burden in lesion sites among all immunized and control animals, whereas only 20% of lesions from rabbit Im1 analysed by darkfield microscopy were positive for motile treponemes as compared with 77.8% and 60% for control rabbits Ct1 and Ct2, respectively. This observation of a

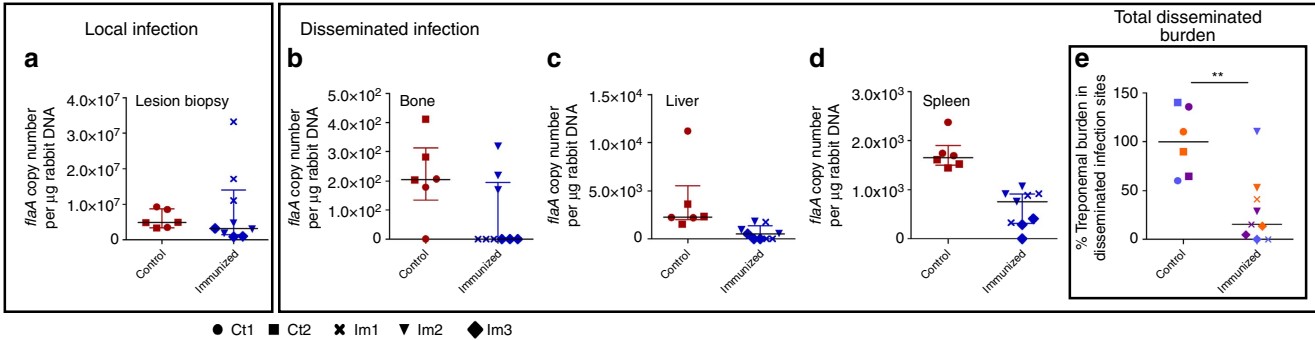

**Figure 3 | Immunization with Tp0751 inhibits *T. pallidum* dissemination.** *T. pallidum* burden was evaluated in control (Ct, *N* = 2) and immunized (Im, *N* = 3) animals using quantitative real-time PCR to measure *flaA* DNA concentrations in lesion biopsies ((**a**); local infection site) and in disseminated infection sites, (**b**) bone, (**c**) liver and (**d**) spleen. Results were normalized to rabbit gDNA concentration and presented as median ± interquartile range (IQR); points correspond to separately extracted samples from each animal. (**e**) Total disseminated burden was assessed between control animals (*n* = 6 samples, 3 samples per animal) and immunized animals (*n* = 9 samples) across all disseminated sites (bone, blue; liver, orange; spleen, purple) using Mann–Whitney test (**P* < 0.005).

| | Donor animal* | Recipient animal† | Test | | |
|---|---|---|---|---|---|
| | | | Seroconversion | Orchitis | Darkfield |
| Control | Ct1 | R-Ct1 | + (Day 31) | + (Day 39) | + |
| | Ct2 | R-Ct2 | + (Day 31) | + (Day 42) | + |
| Immunized | Im1 | R-Im1 | − | − | − |
| | Im2 | R-Im2 | − | − | − |
| | Im3 | R-Im3 | + (Day 73) | − | Not performed |

**Table 2 | Serological and physical markers of testicular infection in RIT rabbits.**

The symbol '+' indicates positive seroconversion (reactive VDRL (≥1:1) and FTA-ABS (≥3+)). The symbol '−' indicates negative seroconversion (nonreactive/weakly reactive VDRL and FTA-ABS (≤2+)).
*Rabbits dermally infected with *T. pallidum*.
†Rabbits that received testicular injections of popliteal lymph nodes from *T. pallidum*-challenged animals.

higher treponemal burden in Im1 could suggest that this animal experienced increased *T. pallidum* retention at the challenge site, and by extension reduced *T. pallidum* dissemination from the primary lesion site. However, these results must be interpreted with caution as it is difficult to reach a definitive conclusion concerning the true lesion burden in Im1 because of the inter-rabbit variability observed that is further confounded by the fact that qPCR is capable of detecting DNA from both live and dead organisms.

To confirm that sterilizing immunity has been achieved, animals receiving lymph nodes from immunized challenged animals must not display any physical or serological signs of infection. Recipient animals of lymph nodes from Im1 and Im2 (R-Im1 and R-Im2) did not seroconvert or develop asymptomatic orchitis post transfer over a 185-day observation period, and darkfield analysis of testes extracts did not detect any presence of *T. pallidum* (Fig. 5b). Immunized rabbit Im3 harboured the lowest treponemal burden at both challenge and distal organ sites and displayed the highest levels of immune cell infiltration at day 14 PC. This high degree of clearance is at odds with the observation that rabbit R-Im3, which received popliteal lymph nodes from rabbit Im3, seroconverted during the course of the experiment, although at a late time point. This seroconversion, which displayed weak VDRL reactivity and did not increase substantially or rapidly for the FTA-ABS assay, may point to the transfer of residual dead treponemes draining to the popliteal lymph nodes in rabbit Im3, and thus may not be indicative of an active infection in this recipient animal. Furthermore, the high levels of cellular infiltrate observed in rabbit Im3 at day 14 PC could suggest that this time point represents the peak immune

response within the lesions of this particular animal. During the natural progression of a rabbit primary lesion, peak treponemal clearance coincides with increasing prevalence of macrophages followed by the detection of organisms within these immune cells[35]. This could further support the notion that treponemes were being cleared and drained into popliteal lymph nodes at this time point. However, in the absence of a more complete cellular infiltration time course, it is currently impossible to determine whether this is the case.

Efficient clearance of *T. pallidum* from primary lesion sites relies upon the development of a T-helper type 1 (Th1) cytokine response produced predominantly by CD4+ T cells[36,37], the subsequent activation of resident macrophages and the presence of specific serum against *T. pallidum* to promote opsonophagocytosis of treponemes by macrophages[35,38,39]. Histological analysis of primary lesion sites in control and immunized animals revealed a significant increase in cellular infiltration in Tp0751-immunized animals at day 14 PC. Increased infiltration of differentiated B cells in primary lesions of immunized animals is suggestive of local production of Tp0751-specific antibodies secreted by these immune cells. These antibodies could be inhibitory or opsonic, thereby facilitating (1) *T. pallidum* retention at the primary lesion sites by inhibiting treponemal dissemination via blocking Tp0751-mediated patho-gen interactions with the extracellular matrix and endothelium or (2) treponemal clearance through opsonophagocytosis, respectively. It is likely that both of these proposed mechanisms contributed to the observed protection against *T. pallidum* dissemination (Fig. 5b), as Tp0751-specific serum has previously been shown to promote *T. pallidum* uptake by rabbit

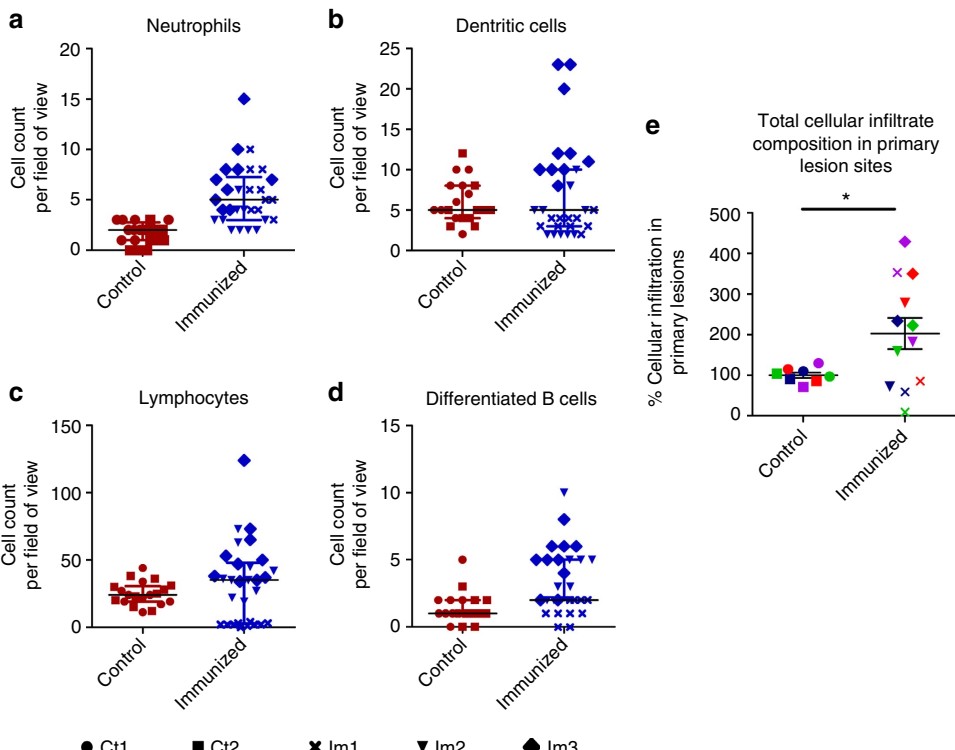

**Figure 4 | Tp0751-immunized animals have increased cellular infiltration in lesions.** Lesion biopsies from control (Ct, $N = 2$) and immunized (Im, $N = 3$) animals on day 14 PC were microscopically evaluated to quantify (**a**) neutrophils, (**b**) dendritic cells, (**c**) lymphocytes and (**d**) differentiated B cells from five fields of view (FOV, $\times 400$) using two vertical sections per biopsy. Shown is the median ± interquartile range (IQR); individual points represent one FOV. (**e**) Total cellular infiltrate composition in lesions (mean ± s.e.) was assessed between control animals ($n = 6$ samples, 3 samples per animal) and immunized animals ($n = 9$ samples) across all cell types (green, lymphocytes; purple, neutrophils; red, differentiated B cells; blue, dendritic cells) using a two-tailed $t$-test (*$P < 0.05$).

peritoneal macrophages[13] and to inhibit *T. pallidum* adhesion to laminin[40]. Primary lesions of immunized animals also contained higher levels of dendritic cells and lymphocytes. Our analyses did not evaluate cytokine secretion or assess activation state or phagocytic activity of the cells. However, the relative abundance of these cell types, taken together with the partial protection against dissemination observed in the study, suggests that an efficient immune reaction was mounted against *T. pallidum* at the local infection site. Such an immune reaction would require participation from lymphocytes for Th1 cytokine production and phagocytic activity by resident macrophages. Higher numbers of neutrophils were also observed in the primary lesions of Tp0751-immunzed animals. Interestingly, neutrophils are typically associated with increased ulceration of primary lesions[41], but all immunized animals displayed consistently lower levels of ulceration throughout the experiment as compared with the unimmunized controls. Neutrophils are prominent early responders during bacterial infections, but are not known to mediate efficient clearance of *T. pallidum*[41]. Thus, the increased prevalence of neutrophils in the lesions of immunized animals could signify effective immune cell homing to the site of infection that would also contribute to bacterial clearance. Furthermore, differences in cellular infiltration between individual immunized animals can be partially attributed to the outbred nature of this animal model. It is also important to consider that histological analyses in this study were performed at a single time point and therefore do not necessarily represent the dynamic nature of the cellular infiltration that occurs in a primary lesion.

This study shows promising findings regarding the effectiveness of Tp0751 as a syphilis vaccine candidate. It should be noted that in this study statistical power is limited because of the small sample size ($N = 2$, control; $N = 3$, immunized) as well as the outbred nature of the rabbit animal model. However, the findings from this study can be used to inform the parameters of future investigations that will include power analyses to determine adequate sample size. Although immunization with Tp0751 did not result in sterile protection within disseminated infection sites of the immunized animals, the sterile protection achieved in animals that received lymph nodes from Tp0751-immunized animals is a positive indication that optimizing vaccine administration, to achieve appropriate antigen and adjuvant doses, could result in sterile immunity in immunized animals. Furthermore, in these future studies we will address whether Tp0751 vaccination results in cross-protection against heterologous *T. pallidum* strains. Notably, the Tp0751 amino-acid sequences are identical in all sequenced *T. pallidum* strains, suggesting this protein will indeed elicit broad protective capacity.

Future studies will also include testing the efficacy of a multicomponent vaccine consisting of Tp0751 and TprK, another promising syphilis vaccine candidate that plays a central role in facilitating *T. pallidum* immune evasion through antigenic variation[42–46]. Previous studies have shown that animals immunized with the N-terminal portion of TprK display attenuated lesion development and smaller treponemal burdens in primary lesions relative to nonimmunized controls[19]. In addition, T cells exhibit reactivity to epitopes found in conserved regions of TprK[47] that may promote Th1 cytokine secretion by T cells to activate macrophages and facilitate treponemal clearance. Formulation of a multicomponent vaccine cocktail using these two protective antigens may enable a two-pronged protective response against both lesion development and treponemal dissemination from the primary site of infection.

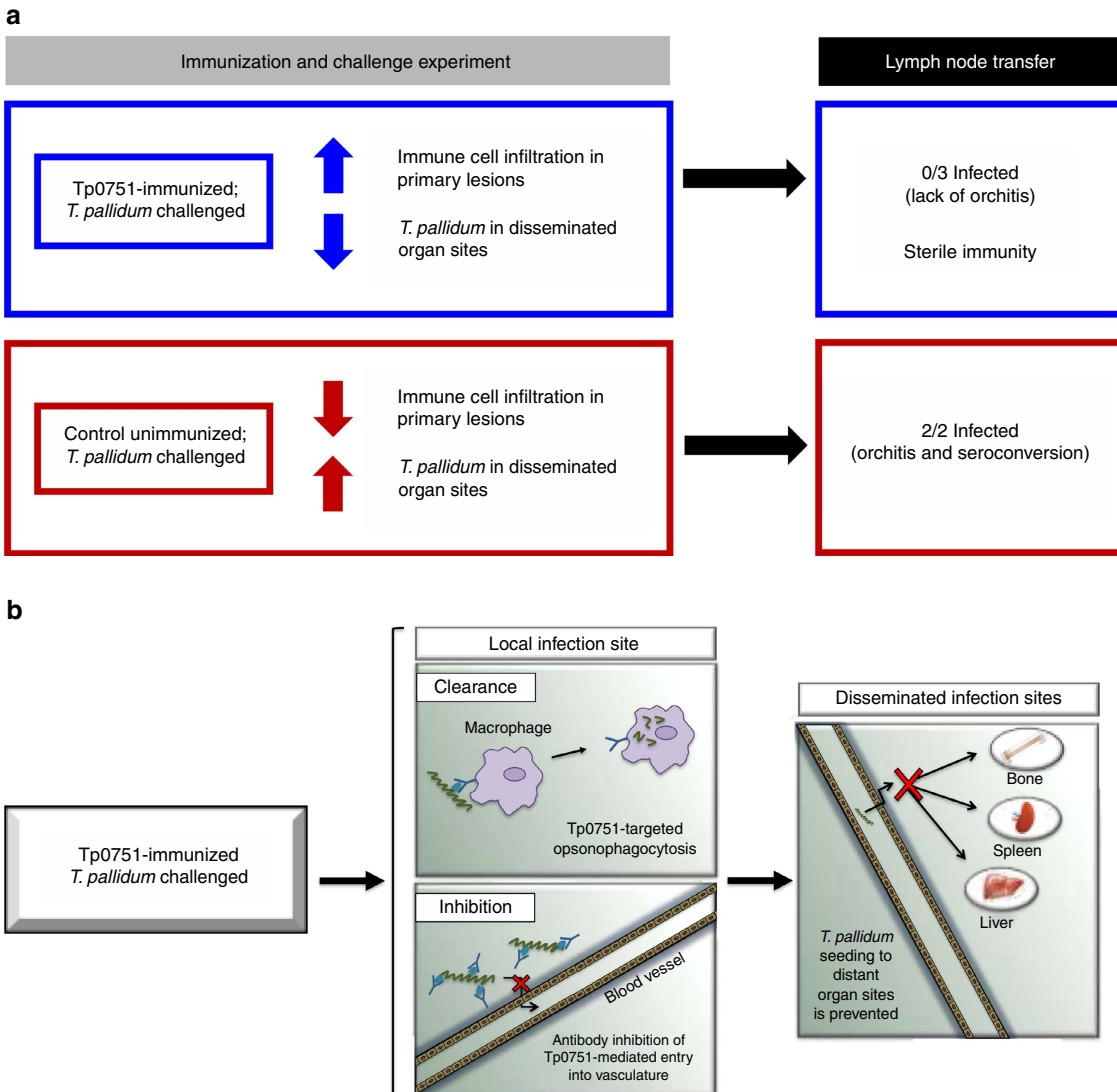

**Figure 5 | Summary of key findings and proposed mechanisms.** (**a**) Summary of key findings. Tp0751 immunization promotes cellular infiltration into primary lesions and inhibits *T. pallidum* dissemination to distant organ sites. Lymph node transfer from Tp0751-immunized to naïve animals does not cause *T. pallidum* infection, confirming sterile immunity. (**b**) Proposed mechanism for the protection against *T. pallidum* dissemination conferred by Tp0751 immunization. We propose that Tp0751 immunization induces specific antibody production in primary lesions, facilitating *T. pallidum* clearance via macrophage opsonophagocytosis. Local Tp0751-specific antibodies would also inhibit *T. pallidum* bloodstream entry by preventing Tp0751-mediated adhesion to blood vessel components. Collectively, these reactions would prevent *T. pallidum* from accessing the bloodstream and secondary infection sites.

Vascular dissemination is a key component of *T. pallidum* pathogenesis, and therefore identifying vaccine candidates involved in this process is critical to achieving successful protection. Evaluation of treponemal burden in the liver, bone and spleen using qPCR revealed that Tp0751 immunization inhibits *T. pallidum* invasion into distant tissue sites. Importantly, the most substantial decrease in *T. pallidum* DNA concentration between immunized and control animals was observed in the spleen, a highly vascularized site that was previously shown to harbour the highest burden of *T. pallidum* during early infection[41]. Previous investigations have implicated Tp0751 in *T. pallidum* dissemination based upon interactions with host components of the vasculature including fibronectin, found at the luminal face of the vascular endothelium, as well as laminin and collagen[15], located in the subendothelial portion of the vasculature. Furthermore, Tp0751 is capable of mediating spirochete attachment to endothelial cells when heterologously expressed by a noninfectious *Borrelia burgdorferi* strain[48]. Recent

structural analysis of Tp0751 has revealed that the C terminus adopts an eight-stranded antiparallel β-barrel lipocalin fold and that protein regions localized to this fold are responsible for interaction with extracellular matrix components, whereas a more discrete region of the lipocalin fold mediates adhesion to endothelial cells[48]. The Tp0751 lipocalin fold is divergent among this family of proteins, but shares important structural similarities with the *Neisseria meningitidis* complement evasion protein, fHbp[49]. Tp0751 and fHbp are both lipidated, surface-exposed lipocalins that are implicated in bacterial pathogenesis[49] and lack the characteristic hydrophobic binding pocket found in the majority of lipocalins. Furthermore, host-binding regions located within these proteins both map along a single face of the lipocalin fold[48]. Of particular relevance to the current study is the fact that fHbp was identified as a promising vaccine candidate for meningococcal serogroup B and is currently part of the multicomponent meningitis vaccines 4CMenB (Bexsero) and rLP2086 (Trumenba). rLP2086 has been successfully tested in

seven clinical studies comprising more than 4,000 human subjects[50]. Bexsero was approved in 2013 by the European Medicines Agency for use in all age groups, and Bexsero and Trumenba were approved in 2015 and 2014, respectively, by the United States Food and Drug Administration for use in individuals 10–25 years of age[51].

The results presented in our study indicate that the *T. pallidum* lipocalin domain-containing Tp0751 is a promising vaccine candidate, as immunization with this protein significantly reduces treponemal dissemination within the host (Fig. 5a). Elimination of dissemination is a central requirement for a syphilis vaccine; this will simultaneously (1) reduce the incidence of syphilis by preventing the formation of the highly infectious secondary lesions characteristic of syphilis, (2) eliminate the devastating consequences of congenital infections, and (3) avert the multitude of serious sequelae associated with syphilis, including neurosyphilis and ocular syphilis. This subunit vaccine candidate, inducing sterile protection against *T. pallidum* infection, may offer a timely solution to the increasing threat of syphilis that is being observed worldwide.

## Methods

**Ethics statement.** All animal studies were approved by the local institutional review board at the University of Victoria, and were conducted in strict accordance with standard accepted principles as set forth by the Canadian Council on Animal Care, National Institutes of Health and the United States Department of Agriculture in a facility accredited by the Canadian Council on Animal Care and the American Association for the Accreditation of Laboratory Animal Care.

**Recombinant protein expression and purification.** *tp0751* (encoding amino acids Cys24-Pro237, with an N-terminal hexahistidine tag) was cloned into the expression vector pDEST17 (Gateway technology, Invitrogen, Burlington, ON, Canada) as previously described[52]. Expression, lysis and purification of soluble Tp0751 (Cys24-Pro237) was performed as previously described[52]. N-terminal amino-acid sequencing of the purified protein, as reported previously[52], confirmed that all SDS-PAGE protein bands present corresponded to full-length Tp0751 and truncated versions of Tp0751 that are reproducibly formed during the purification process (Supplementary Fig. 1).

**Immunization procedure.** We used a cohort of five male SPF New Zealand White rabbits (3.5 kg, 13–15 weeks of age, Charles River Laboratories, Ontario, Canada) with negative VDRL and FTA-Abs serology (Supplementary Table 1). Three animals, randomly selected from the cohort of five rabbits, were sedated and immunized three separate times with water in oil emulsions made up of 0.52 mg ml$^{-1}$ Tp0751 in aqueous buffer with TiterMax Gold adjuvant (Sigma, St Louis, MO, USA) in a 1:1 ratio. Each immunization was delivered as four 0.1 ml subcutaneous injections into the shoulders and two 0.04 ml intramuscular injections into the quadricep muscles, according to the manufacturer's instructions. Rabbits were shaved across the shoulders and injection sites were cleansed with 70% ethanol before injections. The additional immunizations were delivered to each rabbit at 3 and 6 weeks following the initial injection, and this immunization regimen was selected to produce robust antibody responses (Fig. 1).

**T. pallidum propagation.** Propagation of *T. pallidum* subs*p. pallidum*, Nichols strain[53] (hereafter referred to as *T. pallidum*) was performed as per Lukehart and Marra[54], with the exception that testicular extractions were performed under anaerobic conditions in a Coy Laboratory Products anaerobic chamber (Mandel Scientific Company Inc., Guelph, ON, Canada) to enhance *T. pallidum* viability.

**Challenge procedure.** Naïve control rabbits were sedated and bled 10 days before challenge to confirm negative VDRL reactivity (Supplementary Table 2). At 25 days after the final immunization and 10 days after control animals were bled, the rabbit's backs were shaved, cleansed with 70% ethanol and injected intradermally with 0.1 ml of $1 \times 10^7$ *T. pallidum* subsp. *pallidum* (Nichols strain) per ml in 0.9% saline, in each of 10 spots. Challenge sites were monitored daily for erythema, induration and painless ulceration and were measured daily to assess lesion diameter. Challenge sites from control and immunized animals were evaluated for the presence of motile *T. pallidum* 10 days PC by syringe aspiration using a 26-gauge needle; aspirates were examined in a blinded manner for the presence of *T. pallidum* using a Nikon Eclipse E600 darkfield microscope (Nikon Canada, Mississauga, ON, Canada).

**Rabbit infectivity test (RIT).** On day 14 PC, rabbits were euthanized and popliteal lymph nodes were removed, extracted in saline and injected into the testes of randomly assigned naïve anaesthetized rabbits according to the methods of Lukehart and Marra[54]. Animals receiving lymph nodes from the *T. pallidum*-challenged control and immunized animals were monitored daily for visual signs of infection (that is, asymptomatic orchitis) and assayed at least every 3 weeks for seroconversion; animals were immediately euthanized upon orchitis development. Animals that did not develop orchitis were retained until the termination of the experiment at day 185; upon euthanasia, darkfield analysis was performed on testicular extracts from the RIT rabbits to look for evidence of viable *T. pallidum*. Rabbit Im3 was euthanized on day 134 for an unrelated health issue (dietary intolerance diarrhoea); no darkfield analyses were performed on this animal at the time of euthanasia. For further details, please refer to Table 2 and Supplementary Table 2.

**Serological analyses.** Blood samples were obtained from sedated immunized and RIT rabbits via ear bleeds at 3-week intervals; serum was stored at − 20 °C until tested. VDRL slide flocculation and FTA-ABS analyses were performed blindly and as described in the Manual of Tests for Syphilis[55]. The FTA-ABS test was performed using FTA-ABS IFA Test system (Zeus Scientific, Branchburg, NJ, USA) and goat anti-rabbit FITC IgG antibody (Sigma-Aldrich Canada Ltd, Oakville, ON, Canada). The standard 1 + to 4 + FTA-ABS scale was used, where 1 + denotes the lowest reactivity and 4 + denotes the highest reactivity. Antibody titres against the antigen, Tp0751 (Cys24-Pro237), were obtained for the immunized rabbits (Supplementary Fig. 2) using an enzyme-linked immunosorbent assay format. The 96-well NUNC Maxisorp plates (Thermo Fisher Scientific, Ottawa, ON, Canada) were coated with 0.1 μg of recombinant Tp0751 (Cys24-Pro237), and blocked with 1% bovine serum albumin. Rabbit serum was diluted in 1% bovine serum albumin up to 1:20,000. Goat anti-rabbit IgG horseradish peroxidase (Sigma-Aldrich Canada Ltd) was used as a secondary antibody in a 1:3,000 dilution, and binding was detected by using TMB (3,3′,5,5′-Tetramethylbenzidine) peroxidase substrate (KPL, Guelph, ON, Canada) and detecting absorbance at 600 nm with a Biotek Synergy HT (Fisher Scientific, Ottawa, ON, Canada) plate reader. Serum reactivity for Im26, a Tp0751-immunized rabbit used for antibody production, was used as a standard; each individual rabbit was compared with Im26 to account for interexperiment variability.

**Extraction and purification of T. pallidum DNA from tissues.** Genomic DNA (gDNA) extraction was performed on tissues collected from *T. pallidum*-challenged rabbits using a method adapted from the Qiagen QIAamp DNA Mini Kit (Valencia, CA, USA). Lysis Buffer (180 μl; 10 mM Tris pH 8.0, 0.1 M EDTA, 0.5% SDS) was added to ∼25 mg tissue (∼10 mg for spleen) and homogenized using a Qiagen TissueLyser LT operating for 40 s at 30 Hz. Proteinase K (5 mg) was added and digestion was performed overnight at 56 °C. Samples were then treated with RNase A (400 μg) followed by incubation at 70 °C for 10 min. Three tissue samples from each rabbit organ were analysed for reproducibility. Bone gDNA was extracted using the Qiagen QIAamp DNA Micro Kit according to the manufacturer's instructions. Tissue and bone extractions were eluted in 100 and 35 μl, respectively, of Gibco PCR Grade Distilled Water (Thermo Fisher Scientific) and stored at − 80 °C. DNA was quantified using a Beckman Coulter DU 730 Life Science UV/Vis Spectrophotometer (Beckman Coulter Canada, Mississauga, ON, Canada).

**Quantitative PCR.** Quantitative real-time PCR was performed on gDNA extractions of *T. pallidum*-challenged rabbit tissues using a SYBR green I assay. Quantification of *T. pallidum* gDNA was determined using primers targeting a 285 bp region of the endoflagellar sheath protein (*flaA*) gene (GenBank accession number M63142). The sense primer (5′-AACGCAAACGCAATGATAAA-3′) annealed to bases 475 to 494, and the antisense primer (5′-CCAGGAGTCGAACA GGAGATAC-3′) annealed to bases 738 to 759 of *flaA*. Quantification of rabbit gDNA was performed with primers targeting a 267 bp region of exon 1 of the collagenase-1 precursor (*MMP-1*) gene (GenBank accession number M17820). The sense primer (5′- TTGCTTCTTCACACCAGAATGCTGT-3′) annealed to bases 300 to 324 and the antisense primer (5′-GCGTGATCAGGCACTATGTAGC AAT-3′) annealed to bases 542 to 566. All primers were ordered from Integrated DNA Technologies (Coralville, IA, USA). All reactions (20 μl) were performed in quadruplicate and contained in-house SYBR buffer (10 mM Tris pH 8.3, 20 mM KCl, 5 mM (NH$_4$)$_2$SO$_4$, 0.8% glycerol, 0.01% Tween-20, 0.25X SYBR Green I), 0.6 U Fermentas Maxima Hot Start Taq Polymerase (Thermo Fisher Scientific), 200 μM Fermentas dNTPs, 4 mM MgCl$_2$ and 350 nM (*MMP-1*) or 700 nM (*flaA*) primers. The amount of gDNA isolated was variable between tissue types, but was normalized within each tissue type based on the lowest gDNA concentration obtained from spectrophotometric measurements. A standard curve was created for *flaA* using a 10-fold serial dilution from $10^7$ to $10^1$ copies of linearized plasmid DNA with an efficiency of 99.3% and an $R^2$ value of 0.9929. A standard curve was created for *MMP-1* using a twofold serial dilution of rabbit gDNA from 100 to 0.0488 ng μl$^{-1}$ with an efficiency of 99.0% and an $R^2$ value of 0.9924. Assays were run on an Eppendorf Mastercycler ep realplex-4 real-time PCR thermocycler (Mississauga, ON, Canada) using Eppendorf twin.tec white skirted 96-well plates

sealed with Eppendorf Masterclear adhesive real-time PCR Sealing Film and an Eppendorf Heat Sealer. PCR conditions for *flaA* were as follows: 95 °C for 8 min, followed by 40 cycles of 95 °C for 15 s, 65 °C for 20 s, 72 °C for 30 s and an additional extension step at 84 °C for 10 s, after which a final denaturation step of 15 s at 95 °C and a melting curve from 60 °C to 95 °C was performed over 20 min. PCR conditions for *MMP-1* were as follows: 95 °C for 10 min, followed by 40 cycles of 95 °C for 15 s, 64 °C for 20 s and 72 °C for 40 s, after which a final denaturation step of 15 s at 95 °C and melting curve analysis from 60 °C to 95 °C was performed over 20 min. Each assay was run with four controls: (1) no template control; (2) no amplification control (no *Taq* polymerase); (3) no primer control; and (4) positive control with a known concentration or copy number of rabbit gDNA or linearized *flaA* plasmid DNA, respectively.

**Histology.** Biopsy punch samples (2 mm) were taken from a single lesion from each rabbit at 14 days PC and sent to the University of British Columbia Animal Care Service Veterinary Pathologist. Samples were fixed in formalin, mounted, bisected and vertically sectioned at 4 μm. Samples were processed with haematoxylin and eosin staining and CD26 immunohistochemistry for analysis via microscopy to determine abundance of immune cells including lymphocytes, neutrophils, dendritic cells and differentiated B cells. Each cell type was counted blinded at × 400 magnification from five random fields of view from two separate sections. Steiner silver staining was used in an attempt to detect *T. pallidum* (no *T. pallidum* was detected via this staining method).

**Data analysis.** The qPCR threshold cycle ($C_t$) value calculations and melting curve analyses were performed using Eppendorf realplex version 2.2 software; baseline was set automatically with drift correction on and fluorescence thresholds of 393 and 707 for *flaA* and *MMP-1* amplicons, respectively. Data analysis for qPCR was performed using Microsoft Excel 2007 (Microsoft, Mississauga, ON, Canada) and the *flaA* copy was normalized to the amount of rabbit gDNA based upon amplification of *MMP-1*. Data from qPCR and histology experiments were analysed by normalizing the immunized rabbit values between tissue or cell types to the control rabbits by expressing as a percentage relative to the median value of the control rabbits. Sample variance was assessed using *F*-tests and data normality was assessed with a D'Agostino-Pearson omnibus normality test and, based on data variance and normality results, either Mann–Whitney nonparametric test (Fig. 3e) or Student's two-tailed *t*-test (Fig. 4e) was used to evaluate statistical significance. Nonlinear regression with an extra sum of squares *F*-test was used to compare lesion ulceration (Fig. 2a) and diameter (Fig. 2b) between control and immunized rabbits by analysing the difference in slopes using average values between rabbits. *T. pallidum* lesion burden, determined by darkfield analysis (Fig. 2c), was analysed using Student's *t*-test to compare the average lesion burdens between control and immunized rabbits ($P = 0.0287$) and the nonparametric Mann–Whitney test to compare all lesions between control and immunized rabbits ($P = 0.0002$). Statistics were performed, and graphs were constructed, using GraphPad Prism version 7.01 (GraphPad, La Jolla, CA, USA).

**Data availability.** The authors declare that the data supporting the findings of this study are available within the article and its Supplementary Information files.

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

## Acknowledgements

We thank Dr Tara Moriarty, University of Toronto, for assistance with statistical analyses, Brigette Church, University of Victoria, for review of the manuscript, Dr Sheila Lukehart, University of Washington, for critical review of the manuscript and Dr Sheila Lukehart and Barbara Molini, University of Washington, for training on propagation, immunization and challenge procedures as well as lymph node transfer methodology. This work was funded by a Public Health Service Grant from the National Institute of Allergy and Infectious Diseases (NIAID), National Institutes of Health (NIH), to C.E.C under award number R01AI015334. C.E.C. acknowledges the Canada Research Chair program and the Michael Smith Foundation for Health Research Scholar Program for salary support. K.V.L. was supported by a Natural Sciences and Engineering Research Council of Canada Postgraduate Scholarship.

## Author contributions

C.E.C. supervised the study, designed the research, analysed and interpreted data and wrote the manuscript. K.V.L. analysed and interpreted data and wrote the manuscript. R.H. produced the recombinant proteins used for immunizations, performed all the *in vivo* work and analysed and interpreted data. C.W. assisted with the *in vivo* work, and C.W. and D.P. performed all quantitative PCR analyses and analysed and interpreted the results. S.H. analysed and interpreted data. All authors reviewed/edited the manuscript. The funders had no role in study design, data collection and analysis, decision to publish or preparation of the manuscript.

## Additional information

**Competing financial interests:** The authors declare no competing financial interests.

