## [Peer Review File · Nature Communications]

Reviewers' comments:

Reviewer #1 (Remarks to the Author):

A. Summary of the key results

The authors describe a new protein candidate (tp0751) for syphilis vaccination. Tp0751 was reported to play an important role in the crossing of vascular endothelium and thus facilitates dissemination of *T. pallidum* via the bloodstream. A rabbit immunization model was used to investigate efficacy of tp0751 vaccination to block *T. pallidum* dissemination in vivo.

B. Originality and interest: if not novel, please give reference

The study is a logical follow up of the previous work of the author(s) and complements nicely the work that was done on the characterization of the lipoprotein tp0751 and its function. Three of the authors hold a relatively new patent (WO 2015004604 A1) on immunogenic tp0751 fragments in which some aspects of the study have been described. The search for suitable targets for syphilis vaccination is of great importance to counteract treponematosi. The study is therefore relevant and provides a significant contribution for the development of a vaccine against syphilis.

C. Data & methodology: validity of approach, quality of data, quality of presentation

Overall the MS reads well and the data are presented nicely. However, parts of the Material and Methods need additional information e.g. it is not even introduced which *T. pallidum* strain has been used to challenge the rabbits. For details see P-2-P comments following point H).

D. Appropriate use of statistics and treatment of uncertainties

Animal numbers and statistics are the major weakness of the study. The authors explanation provided in the